# Engineering a synthetic gene circuit for high-performance inducible expression in mammalian systems

Giuliano De Carluccio [1,2,3], Virginia Fusco [1,2] & Diego di Bernardo [1,2] ✉

Inducible gene expression systems can be used to control the expression of a gene of interest by means of a small-molecule. One of the most common designs involves engineering a small-molecule responsive transcription factor (TF) and its cognate promoter, which often results in a compromise between minimal uninduced background expression (leakiness) and maximal induced expression. Here, we focus on an alternative strategy using quantitative synthetic biology to mitigate leakiness while maintaining high expression, without modifying neither the TF nor the promoter. Through mathematical modelling and experimental validations, we design the CASwitch, a mammalian synthetic gene circuit based on combining two well-known network motifs: the Coherent Feed-Forward Loop (CFFL) and the Mutual Inhibition (MI). The CASwitch combines the CRISPR-Cas endoribonuclease CasRx with the state-of-the-art Tet-On3G inducible gene system to achieve high performances. To demonstrate the potentialities of the CASwitch, we apply it to three different scenarios: enhancing a whole-cell biosensor, controlling expression of a toxic gene and inducible production of Adeno-Associated Virus (AAV) vectors.

Inducible gene expression systems play a crucial role in biological research, gene therapy, and biotechnology manufacturing[1–4]. They typically comprises two main components: a transcription factor (TF), whose activity is controlled by a small molecule, and a cognate promoter harbouring TF-specific DNA-binding sequences[5,6]. The performance of an inducible gene expression system can be described by three main features: Leakiness, Maximum Expression, and Fold Induction, as depicted in Fig. 1a. The leakiness is defined as the basal gene expression in the absence of the inducer; the maximum expression is the gene expression at saturating concentration of the inducer; and the fold induction is defined as the ratio between the maximum expression and the leakiness. A high-performance inducible gene expression system should have low leakiness and high maximal expression thus yielding a high fold induction. However, the design of inducible expression systems that meet these criteria poses significant challenges, as it involves multiple cycles of trial-and-error to engineer the transcription factor and its cognate promoter[7,8] that ultimately lead to compromises between leakiness and maximal expression level[9–12].

Here, we focussed on an alternative strategy using synthetic gene circuits to mitigate leakiness while maximising induced expression, without modifying neither the TF nor the promoter. We first mathematically modelled alternative gene circuit topologies to augment the naïve configuration of an inducible gene expression system, as shown in Fig. 1b, where the transcription factor X directly regulates the target gene Z by means of a cognate promoter. We then set to biologically implement these alternative synthetic circuits in mammalian cells to improve the performance of the state-of-the-art and widely used Tet-On3G inducible gene expression system[13]. To this end, we leveraged the Rfx-Cas13d (CasRx) CRISPR-Cas endoribonuclease's unique properties and developed a powerful albeit simple inducible gene expression system that we named CASwitch.

The CASwitch exhibited greatly improved performances against the state-of-the-art Tet-On3G system in terms of negligible leakiness

[1]Telethon Institute of Genetics and Medicine, Naples, Italy. [2]University of Naples Federico II, Department of Chemical Materials and Industrial Engineering, Naples, Italy. [3]Present address: Institute for Medical Engineering and Science, MIT, Cambridge, MA, USA. ✉e-mail: dibernardo@tigem.it

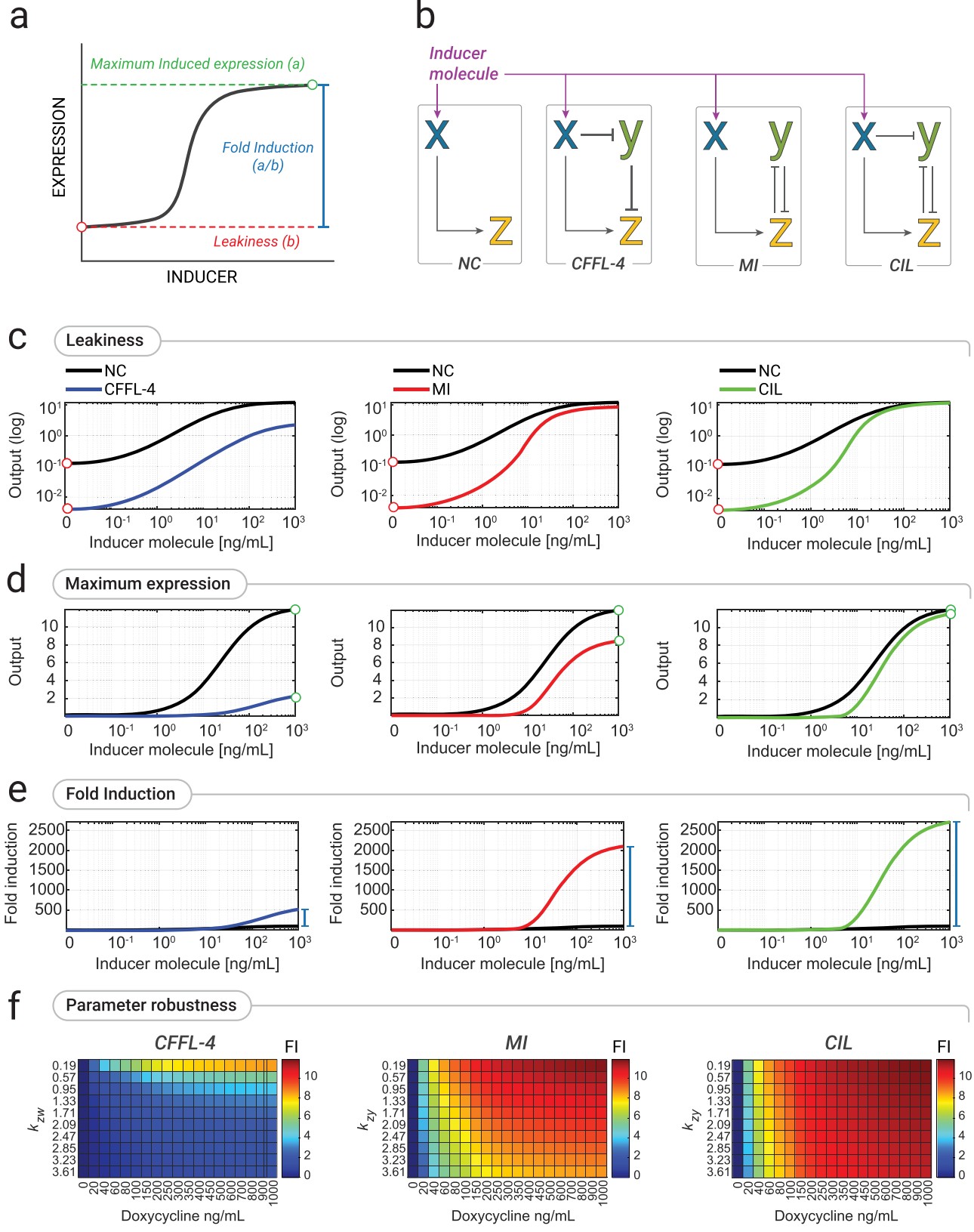

and high maximum expression. We finally showcased the CASwitch platform for two proof-of-principle biotech applications: a whole-cell intracellular copper biosensor with significant improved performances, and for the inducible production of Adeno-Associated Virus (AAV) vector in HEK293T cells.

## Results

### In silico design and analysis of synthetic circuits for high-performance inducible gene expression

Among the potential gene network motifs, we focussed on those that may yield reduced leakiness levels[14]. We thus mathematically modelled

**Fig. 1 | Alternative synthetic circuits for high-performance inducible gene expression. a** The dose response curve of an inducible gene expression system and its features. Leakiness: basal gene expression level in the absence of the inducer. Maximum expression: gene expression at saturating concentration of the inducer. Fold Induction: ratio between Maximum expression and Leakiness. **b** Schematics of the naïve configuration (NC) of an inducible gene expression system and of alternative circuits: CFFL-4, Coherent Feed Forward Loop 4; MI, Mutual Inhibition; CIL, Coherent Inhibitory Loop. Pointed arrows represent activation while blunt head arrows represent repression. The activity of species X can be modulated by an inducer molecule. **c**–**e** Numerical simulations of the level of species Z at different concentrations of the inducer molecules for the different circuits: CFFL-4 (blue), MI (red), and CIL (green) against the naïve TF (black). Simulated output values are shown either in a log-scale (top) to highlight differences in the absence of inducer molecule (leakiness), or as a linear-scale (middle) to highlight difference at the highest concentration of the inducer molecule (maximum expression), or as fold-induction values (bottom) computed as the ratio between each output value and its value in the absence of inducer molecule. **f** Numerical analysis of the Fold Induction (FI) in the three circuits for increasing levels of the inducer molecule (denoted as **D**) on the x-axis and for different numerical values of the parameter $K_{zy}$ representing the repression strength of species Y over species Z. TF: Transcription-based inducible gene system; CFFL-4: Coherent Feed Forward Loop-4 circuit topology; MI: Mutually Inhibition circuit topology; CIL: Coherent Inhibitory Loop circuit topology.

and compared three alternative circuit topologies for inducible gene expression as shown in Fig. 1b, against the naïve configuration (NC): (i) the coherent feedforward loop type 4 (CFFL-4)[15]; (ii) the mutual inhibition (MI) topology[14]; and (iii) a combination of these two topologies that we named Coherent Inhibitory Loop (CIL). All these circuits make use of an additional species Y to inhibit the reporter gene Z in the absence of the inducer molecule, thereby suppressing leaky expression. We used ordinary differential equations and dynamical systems theory to analyse the performance of these three networks, assuming realistic biological parts (Supplementary Note 1).

Analytical results and numerical simulations of the circuits, when using the very same parameters for the common biological parts, confirmed that all three exhibit improved performances over the naïve configuration, in terms of lower leakiness, high maximum expression, and increased fold induction, as reported in Fig. 1c-e and Supplementary Note 1, albeit with notable differences. In the CFFL-4, the leakiness is smaller than the one of the NC thanks to the inhibitory action of Y over Z, in the absence of the inducer molecule (Fig. 1c); however, as X does not fully repress Y upon inducer molecule treatment, the maximal expression of Z is also smaller (Fig. 1d), thus leading to only a modest increase in Fold Induction (Fig. 1e). The MI improves on the CFFL-4 in terms of maximum expression (Fig. 1d), as Y is now repressed also by Z in addition to X. The CIL combines the advantages of both circuits, and it exhibits the best performance as compared to the NC configuration in terms of all the three features, as shown in Fig. 1c-e. To further explore the robustness of these findings, we conducted additional numerical simulations by varying the model's parameters, whose results are shown in Fig. 1f and Supplementary Note 1. For all the parameter values tested, the CIL circuit exhibited the best performance whereas the CFFL-4 was the worst. Based on these analyses, we decided not to biologically implement a CFFL-4 system and instead focused on the biological implementation of the MI and CIL circuits.

## Experimental implementation of the mutual inhibition (MI) circuit by means of pre-gRNA processing of the CasRx endoribonuclease

To experimentally implement mutual inhibition (Fig. 2a), we looked for a biological implementation that was compact and could be applied to any gene of interest Z. We thus turned to CRISPR-Cas endoribonucleases which have been recently repurposed to act as post-transcriptional regulators by exploiting their pre-gRNA processing mechanisms[16]. Indeed, CRISPR-endoribonucleases can cleave specific short sequences known as direct repeats (DRs) on their cognate pre-gRNAs, generating shorter guide RNA (gRNA) sequences; hence, these DRs have been repurposed as cleavage motifs to stabilize or degrade user-defined mRNA transcripts by placing them in the mRNA untranslated regions (UTRs)[16]. Specifically, in our implementation shown in Fig. 2a, we employed the CasRx endoribonuclease to implement species "Y", while species "Z" is the Gaussia Luciferase (gLuc) reporter gene bearing the DR sequence in its 3'UTR. Because of the CasRx's distinctive feature of irreversibly bind its processed gRNA[17], we

reasoned that this configuration could implement a mutual inhibition between species Y and Z. Here, Y is able to negatively regulate Z, as the CasRx cleaves the DR in the 3'UTR of the gLuc mRNA thus leading to the loss of its polyA tail and subsequent degradation; at the same time, we assumed that Z could be able to inhibit Y by "sponging out" the CasRx, which irreversibly binds to the DR and it is thus unable to cleave additional Z mRNAs.

To experimentally test this hypothesis, we co-transfected HEK293T cells with CasRx along with one of three different gLuc transcript variants, as reported in Fig. 2b. These variants bear different numbers of DR motifs at their 3' UTR: either no DR motif, one DR motif, or four DR motifs (4xDR). Our rationale was that by introducing more than one DR motif, we could "sponge" CasRx more effectively and thus alleviate repression of the target gLuc mRNA. Indeed in this scenario, one gLuc-4xDR mRNA should able to bind four CasRx, rather than only one, as in the case of the gLuc-DR. Results are shown in Fig. 2b: in the absence of CasRx, all the three gLuc transcripts yield the same luciferase expression level, independently of the number of DRs in their 3'UTR, thus excluding perturbations of mRNA stability caused by the DR itself. In the case of the gLuc-DR transcript (with one DR), the relative increase in the amount of co-transfected CasRx resulted in an exponential decrease in luciferase expression, with up to 100-fold reduction in luminescence. On the contrary, for the gLuc-4xDR, the CasRx repression efficiency was strongly reduced, thus supporting the hypothesis of a DR-mediated "sponging" of CasRx, although we cannot exclude alternative mechanisms. Encouraged by these results, we sought to implement the MI circuit using the CasRx endoribonuclease, developing the CASwitch v.1 system, as shown in Fig. 2c.

We chose as species X the tetracycline transactivator (rtTA3G) transcriptional factor, which is a fusion protein that combines a tetracycline-responsive DNA-binding domain with a strong transcriptional activation domain[13]. In the presence of the doxycycline, rtTA3G binds to multiple copies of the tetracycline operon (TO) sequence present in its cognate pTRE3G synthetic promoter, thereby inducing the expression of the downstream gene of interest[18]. In the CASwitch v.1 system, both the CasRx and the rtTA3G are constitutively expressed from the CMV promoter, while the gLuc harbours one DR in its 3'UTR and it is placed downstream of the pTRE3G promoter, as schematically shown in Fig. 2c.

We experimentally compared the performances of the CASwitch v.1 and the Tet-On3G by transiently transfecting HEK293T cells with three plasmids: (1) the pCMV-rtTA3G, (2) the pTRE3G-gLuc-DR for the CASwitch v.1, or the pTRE3G-gLuc for the Tet-On3G, and (3) the pCMV-CasRx at a relative molar ratio of 1:5:1. Observe that for the Tet-On3G system, the gLuc has no DR in its 3'UTR, but we co-transfected the CasRx anyway to exclude potential biases caused by cellular burden. We then quantified gLuc expression by luminescence measurements at varying concentrations of doxycycline. Results are reported in Fig. 2d-e and demonstrate that the CASwitch v.1, in the absence of doxycycline, strongly reduces leaky gene expression by >1-log when compared to the Tet-On3G system (Fig. 2d); at the same time, the maximal

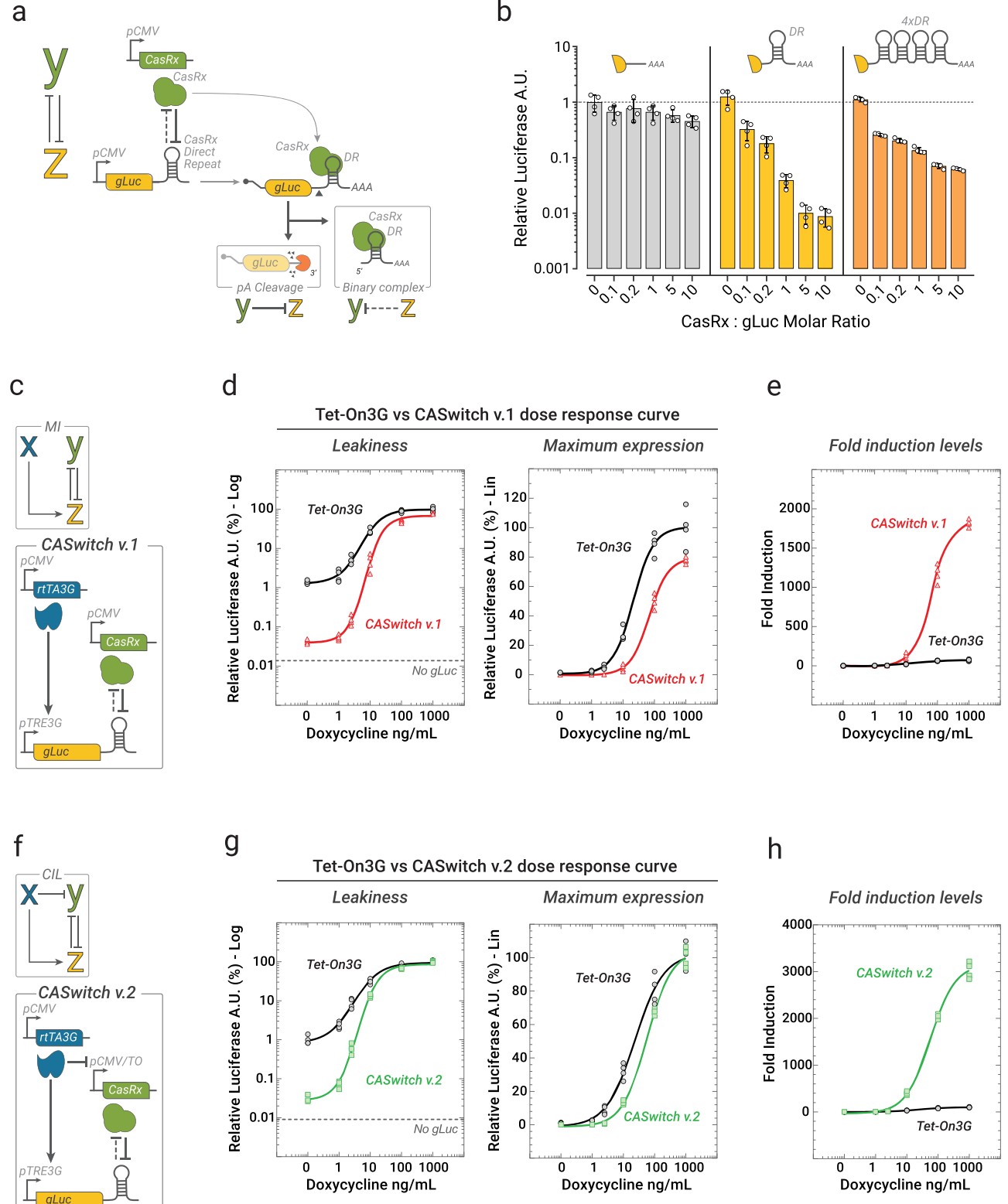

expression upon doxycycline treatment was only slightly reduced (Fig. 2d). Notably, the reduced leakiness and the retention of high maximal expression resulted in a very significant gain in terms of fold-induction by more than 1-log (Fig. 2e).

To further evaluate the robustness of the CASwitch v.1 system, we repeated the same experiments at higher relative concentrations of CasRx, as reported in Supplementary Fig. 1. This resulted in a further suppression of leakiness, but also in a reduction of the maximal

achievable expression, suggesting that controlling CasRx expression is an important design parameter to achieve the desired inducible system properties.

Overall, our results demonstrate that the constitutively expressed CasRx, combined with its cognate direct repeat (DR) in the 3'UTR of a target mRNA, can serve as a plug-and-play strategy to significantly enhance the performance of transcriptional inducible gene expression systems.

**Fig. 2 | The CASwitch inducible gene expression system. a** Experimental implementation of the mutual inhibition. CasRx acts as species Y. The Gaussia Luciferase (gLuc) with a Direct Repeat (DR) in the 3′Untranslated Region (UTR) acts as species Z. CasRx binds to the DR and cleaves the polyA tail (AAA) of the gLuc mRNA leading to its degradation, thus achieving Y-mediated repression of Z. Following cleavage, the CasRx irreversibly binds to the DR forming the gRNA-Cas binary complex which cannot cleave additional mRNAs, thus possibly implementing the Z-mediated repression of Y. **b** Experimental validation of CasRx-mediated mRNA degradation. Cells were transfected with CasRx and gLuc plasmids at the indicated relative concentrations. The bar-plot reports the mean Relative Luciferase in arbitrary units (A.U.) obtained by dividing the average Luciferase A.U. value at each molar ratio by the average Luciferase A.U. value in the absence of CasRx. Error bars correspond to the standard deviation. $n = 4$ biological replicates (white dots). **c** CASwitch v.1.: rtTA3G and CasRx are constitutively expressed from the pCMV promoter, while gLuc with the DR is placed downstream of the pTRE3G promoter. **d, e** Experimental validation of CASwitch v.1 (red) and comparison with the Tet-On3G expression system (black) at the indicated concentrations of doxycycline. $n = 4$ biological replicates. Relative Luciferase A.U. is computed as the Luciferase A.U. value of each data point divided by the average value of the Tet-On3G system at 1000 ng/mL, in both log-scale and linear-scale, and in (**e**) as fold-induction computed as the Luciferase A.U of each data point divided by the average value in the absence of doxycycline. **f** The CASwitch v.2: rtTA3G is constitutively expressed from a pCMV promoter, CasRx is driven by the pCMV/TO that can be repressed by the rtTA3G, while the gLuc with the DR is placed downstream of the pTRE3G promoter. **g, h** Experimental validation of CASwitch v.2 (green) and comparison with the state-of-the-art Tet-On3G gene expression system (black) at the indicated concentrations of doxycycline. $n = 4$ biological replicates. MI: Mutual Inhibition circuit topology; CIL: Coherent Inhibitory Loop circuit topology. Source data are provided as a Source Data file.

## Experimental implementation of the Coherent Inhibitory Loop (CIL) by transcriptional inhibition of CasRx boosts CASwitch performance

We set out to further enhance the performances of the CASwitch v.1 by specifically focusing on the increase in the maximal achievable expression upon doxycycline treatment. To this end, guided by the modelling results in Fig. 1b, we sought to biologically implement the CIL circuit by replacing the constitutive pCMV promoter driving the CasRx with a modified version, named pCMV/TO, as shown in Fig. 2f. The pCMV/TO promoter has two TO sequences downstream of the TATA binding box of the pCMV[19], hence, upon doxycycline administration, rtTA3G binds to these elements and causes a steric hindrance to the PolII resulting in a partial repression of CasRx transcription. We first confirmed the effective doxycycline-dependent inhibition of the pCMV/TO promoter (Supplementary Fig. 2). Subsequently, we verified that switching the pCMV promoter with the pCMV/TO promoter did not affect CasRx expression and its effect on its downstream target (Supplementary Fig. 3). Finally, we proved that the pCMV/TO enables doxycycline-mediated repression of CasRx expression and relief of CasRx-mediated degradation of the target mRNA (Supplementary Fig. 4).

We thus leveraged the pCMV/TO-mediated transcriptional control of the CasRx to implement the CASwitch v.2, as shown in Fig. 2f, and we experimentally compared its performances to that of Tet-On3G system. Results are reported in Fig. 2g,h in terms of luciferase expression at varying concentrations of doxycycline. The CASwitch v.2 exhibited more than 1-log reduction in leakiness compared to the art Tet-On3G system, yielding results similar to those obtained with the CASwitch v.1 (Fig. 2g); this time, however, in agreement with the in silico analysis, it was able to fully recover the maximal achievable expression to the level of the original Tet-On3G (Fig. 2g), thus leading to a very large amplification of fold induction levels of up to 3000-fold (Fig. 2h).

To assess the robustness of CASwitch v.2, we tested its performance against that of state of the art Tet-On3G system by: (i) changing the plasmid molar ratio among the circuit components; (ii) testing it in a different mammalian cell line; and (iii) changing the promoter that drives the rtTA3G.

Results on the performance against changes in plasmid molar ratios are presented in Supplementary Fig. 5. Different amounts of plasmids can affect basal and induced levels of gene expression; hence one may presume that the Tet-On3G system performance could be improved by simply changing the plasmid ratios. Interestingly, the CASwitch v.2 (red and blue lines in Supplementary Fig. 5b,c) maintains its enhanced performance over the Tet-On3G system (yellow and green lines) independently of the plasmid ratio used.

Results on the performance of the CASwitch v.2 in HeLa cells are shown in Supplementary Fig. 6a-b, where it is evident that it retains its improved performance over the Tet-On3G system, consistently with the results observed in HEK293T cells.

Results on the impact of replacing the pCMV promoter driving rtTA3G in the CASwitch v.2 system with two alternative promoters with lower expression strengths (pEF1a and pPGK) are shown in Supplementary Fig. 7. In the cases tested, the CASwitch v.2 exhibited a better performance versus the Tet-On3G system by exhibiting a lower leakiness while maintaining the maximal expression (Supplementary Fig. 7b) thus leading to a higher fold induction (Supplementary Fig. 7c), with a slight decrease in performance for the weakest pPGK promoter. Notably, the use of the pEF1a yielded the highest fold induction, hence we chose to express the rtTA3G from this promoter in following experimental applications of the CASwitch v.2.

Overall, these results confirm that the CASwitch v.2 represents a general strategy to endow transcriptional inducible gene expression system with very low leakiness but with unaltered maximal expression, hence resulting in very large gain in fold induction.

## CASwitch v.2 is an easy-to-implement solution to improve the performance of mammalian whole-cell biosensors

As the CASwitch v.2 greatly enhances the fold induction levels of the Tet-On3G inducible gene expression system, we decided to deploy it to increase the performance of established transcription-based biosensors[20]. As a case in point, we deployed the CASwitch v.2 to improve the performance of a previously published copper biosensor[20] in mammalian cells, as shown in Fig. 3a. In this biosensor, a luciferase reporter gene is placed downstream of a synthetic metal-responsive promoter (pMRE). This promoter is bound by the endogenous metal response element binding transcription factor 1 (MTF-1)[21] in the presence of zinc (Zn), copper (Cu), or cadmium (Cd) driving expression of the downstream reporter gene. As most biosensors, this configuration has several limitations, including low expression of the reporter gene and a narrow dynamic range, defined as the ratio between the maximum achievable biosensor response and its leakiness (Fig. 3b, c−blue line). To address these limitations, we modified the CASwitch v.2 system by replacing the pCMV promoter driving the expression of rtTA3G, with the metal-responsive promoter pMRE, as shown in Fig. 3a, with the goal of simultaneously enhancing the copper biosensor's absolute expression and amplifying its dynamic range.

To evaluate the effectiveness of the CASwitch v.2 plug-in strategy, we compared it to an additional biosensor configuration as shown in Fig. 3a, where the pMRE promoter drives the expression of the rtTA3G transcription factor, which in turn drives expression of fLuc from the pTRE3G promoter. This configuration, in the presence of doxycycline, effectively implements a transcriptional amplification of the reporter gene expression, which however should not improve the dynamic range as both leaky and maximal gene expression should increase.

We evaluated the expression of fLuc from the three configurations at increasing concentrations of copper and at a fixed concentration of doxycycline. Results are reported in Fig. 3b,c: the standard copper biosensor exhibited considerable leakiness and low

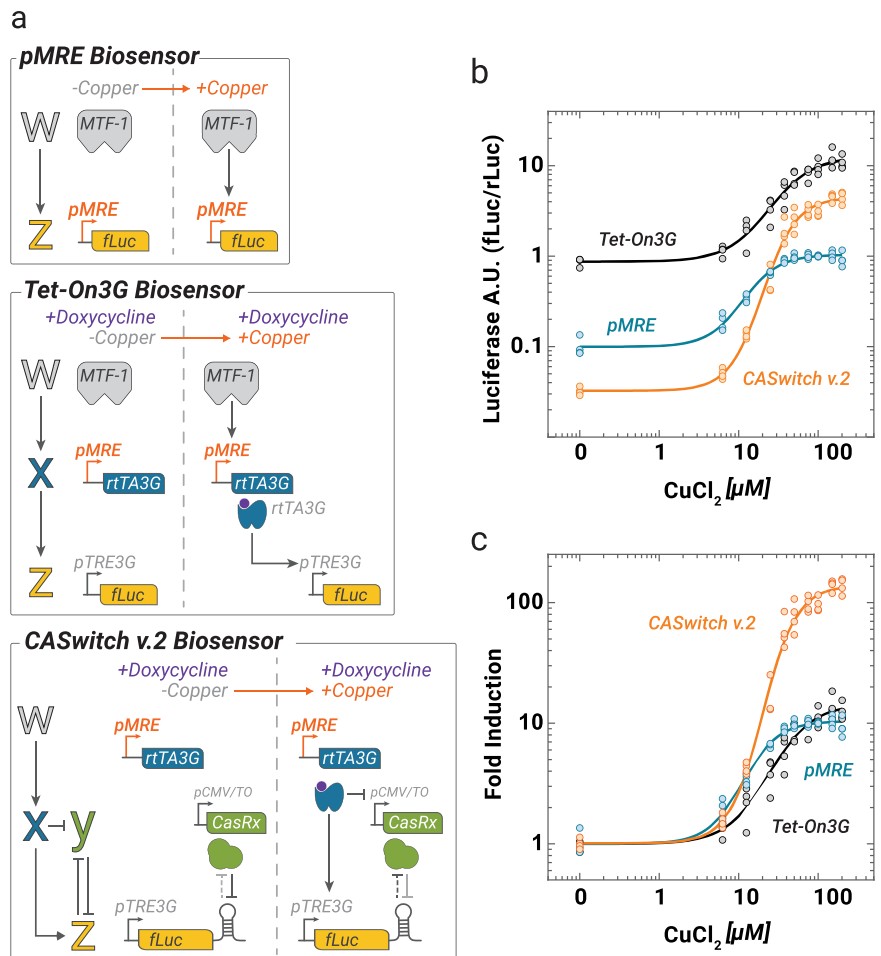

**Fig. 3 | CASwitch v.2 as a whole-cell biosensor. a** Schematics of three alternative experimental implementations of a copper biosensor. Upon copper administration, the endogenous MTF-1 transcription factor binds its cognate synthetic promoter pMRE that either directly drives expression of Firefly Luciferase (fLuc) expression (pMRE Biosensor), or drives expression of the rtTA3G transactivator, which in turn induces the expression of the fLuc through the pTRE3G in the presence of doxycycline (Tet-On3G Biosensor). In the CASwitch v.2 Biosensor, the pMRE promoter drives expression of the rtTA3G, which in turn induces expression of the fLuc harbouring a DR and inhibits expression of the CasRx through the pCMV/TO promoter. **b,c** Experimental validation of the three biosensors at the indicated concentrations of copper chloride in HEK293T cells. Firefly luciferase (fLuc) expression was evaluated by luminescence measurements and normalised to Renilla firefly (rLuc) luminescence. Fold-induction in (**c**) is obtained by dividing each data point by the average luciferase expression in the absence of copper. $n = 4$ biological replicates, albeit for $CuCl_2$ equal to 25uM which shows 3 replicates. MTF-1: metal-responsive transcription factor 1; pMRE: synthetic metal responsive promoter; DR: direct repeat sequence; rtTA3G: reverse tetracycline TransActivator 3G; pTRE3G: Tetracycline Responsive Element promoter 3G; pCMV/TO: modified CMV promoter with two Tetracycline Operon (TO) sequences. Source data are provided as a Source Data file.

levels of reporter gene expression even at high copper concentrations, thus resulting in a low signal-to-noise ratio with a maximum induction of only 10-fold. The second configuration with the rtTA3G resulted in a significant increase in luciferase expression levels at all copper concentrations, however, as expected, it did not lead to dynamic range amplification, as it also increased the leaky reporter expression in the absence of copper. Conversely, the CASwitch v.2 configuration effectively reduced leakiness in the absence of copper, while achieving higher luciferase expression than that of the standard copper biosensor (Fig. 3b). This resulted in a large increase in the biosensor's signal-to-noise ratio with a maximum induction of up to 100-fold, hence 1-log more than the other two configurations (Fig. 3c). Of note, the CASwitch v.2 yielded higher fold-induction levels at four times lower copper concentration, thus also enhancing its sensitivity. Taken together, these findings support the application of the CASwitch v.2 system to improve the efficacy of existing transcriptional-based biosensors that experience limitations in terms of a narrow dynamic range. The expansion of the biosensors' dynamic range through the integration of CASwitch v.2 will yield a more sensitive and reliable biosensor, capable of detecting lower concentrations of the analyte with increased confidence.

## CASwitch v.2 is a tight gene expression system for reliable inducible control of toxic gene expression

We investigated the application of the CASwitch v.2 system in tightly controlling the expression of toxic genes, this feature is very useful for some industrial applications such as recombinant protein production, where the unintended accumulation of the protein of interest due to leakiness impairs host cell viability and lowers production yields (e.g, viral proteins). As a proof-of-principle, we used the CASwitch v.2 system to express the Herpes Simplex Virus Thymidine Kinase-1 (HSV-TK), which exerts cytotoxic effects in the presence of nucleotide analogues such as ganciclovir (GCV)[22]. To this end, as shown in Fig. 4a, we added a Direct Repeat in the 3'UTR of the HSV-TK gene and placed it downstream of the pTRE3G promoter in the CASwitch v.2 circuit. We then evaluated cell viability in the presence of ganciclovir, either with or without doxycycline and compared it to the one obtained by using the state-of-the-art Tet-On3G gene expression system. To account for

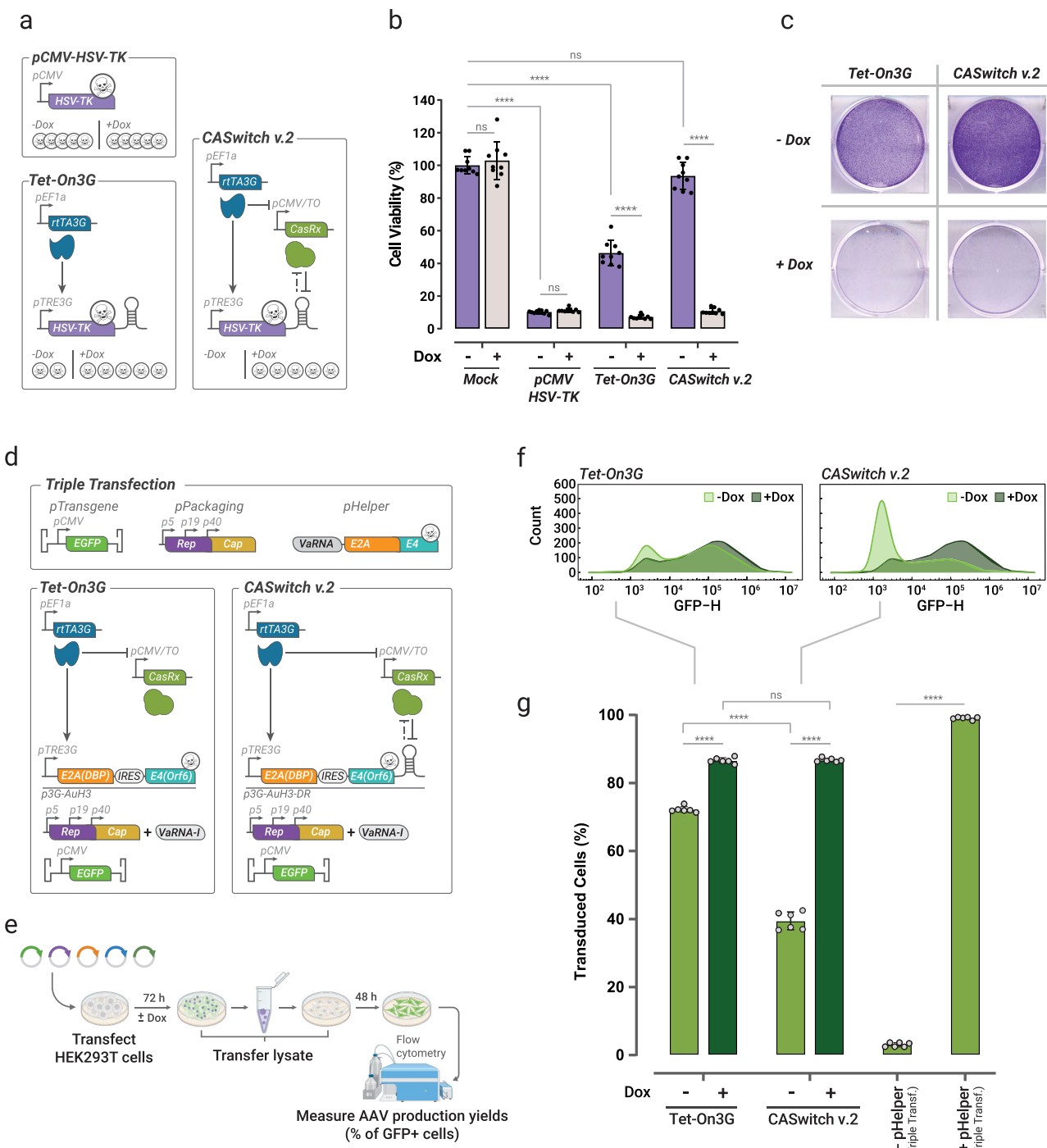

**Fig. 4 | CASwitch v.2 enables inducible expression of toxic genes and inducible production of Adeno Associated Viruses. a** Three alternative constructs to express the cytotoxic HSV-TK gene. pCMV-HSV-TK: positive control, with constitutive expression of HSV-TK. Tet-On3G: the constitutively expressed rtTA3G induces the cytotoxic HSV-TK gene harbouring a DR in its 3'UTR, binding to pTRE3G in the presence of doxycycline. CASwitch v.2: the same as the Tet-On3G but for the presence of the CasRx downstream of the pCMV/TO. **b** Viability of HEK293T cells transfected with the indicated constructs and grown in the presence of ganciclovir. Mock transfected cells represent the negative control. Cell viability is reported as a percentage of the viability of mock transfected cells in the absence of doxycycline. The error bars represent the mean and standard deviation of biological replicates across two independent experiments ($n = 9$). Statistical analysis with ANOVA (one-tailed) after determining equal or unequal variances by D'Agostino & Pearson test (****$P$-value < 0.0001) **c** Crystal violet staining of transfected HEK293T cells to highlight viable cells. **d** Plasmids required for AAV production.

Two alternative experimental implementations for inducible expression of the Helper genes using either the Tet-On3G system or the CASwitch v.2 are also shown. **e** Assay for testing AAV vector inducible production yield by means of viral transduction. Created with Biorender. **f**, **g** Flow cytometry of cells transduced with cell lysates of HEK293T cells transfected with the indicated configurations. At least 10,000 cells were analysed for each point. The bar-plot in (**g**) reports, for each experimental condition, the mean value of the percentage of transduced cells across of biological replicates for two independent experiments ($n = 6$) with error bars corresponding to the standard deviation. Statistical analysis by ANOVA (one-tailed), after determining equal or unequal variances by D'Agostino & Pearson test (****$P$-value < 0.0001). HSV-TK: Herpes Simplex Virus Thymidine Kinase; AAV: Adeno-Associated Virus; E2A(DBP): Early 2A DNA Binding Protein gene; E4(Orf6): Early 4 Open reading frame 6 gene; VaRNA-I: Viral associated RNA-I; Rep: AAV-2 Replication genes; Cap: AAV-2 Capsid genes. Source data are provided as a Source Data file.

cytotoxic effects associated with transfection, we co-transfected cells with a non-coding plasmid in the "Mock" condition, against which all other cell viability measurements were normalized to. Furthermore, constitutive expression of HSV-TK provided a reference for the maximum achievable toxicity. Results are reported in Fig. 4b, c and show no cytotoxic effects for the CASwitch v.2 system in the absence of doxycycline. In contrast, the Tet-On3G system exhibited high cell toxicity, resulting in ~50% cell death in the absence of doxycycline. These findings confirm that the CASwitch v.2 system has very low leakiness, highlighting its efficacy in controlling toxic genes expression.

## CASwitch v.2 as a platform for Adeno-Associated Virus (AAV) production for gene therapy

Adeno-Associated Virus (AAV) vectors have emerged as highly promising tools for in-vivo gene therapy in clinical applications[23]. However, current large-scale industrial bioproduction face challenges in terms of efficiency and scalability, as it mainly relies on transient transfection of HEK293 cell lines[24,25]. Attempts to develop more scalable systems, such as AAV producer cell lines with stable integration of inducible gene systems to control the expression of viral genes, have been hampered by the toxicity associated with leaky expression of viral genes[26–30]. In this context, the CASwitch v.2 expression system may offer a reliable solution having the ability to significantly reduce leakiness while maintaining high levels of maximal achievable expression.

As shown in Fig. 4d, transient triple transfection manufacturing of AAV vectors requires three plasmids: (i) a Transgene plasmid encoding the desired transcriptional unit to be packaged, (ii) a Packaging plasmid, and (iii) a Helper plasmid. The Packaging plasmid in our implementation carries the wild-type AAV2 Rep and Cap genes, while the Helper plasmid contains the E2A, E4, and VA RNAI genes derived from Human Adenovirus 5 (HAdV-5)[31]. As the HAdV-5 genes are polycistronic and expressed from distinct promoters, we first determined the minimal set of viral genes necessary for AAV vector production. Previous studies have shown that the E2A(DBP) and E4(Orf6) coding sequences, along with the VARNA-I ncRNA, are essential for AAV vector production[32]. Therefore, we designed constructs expressing E2A(DBP) and E4(Orf6) as a single transcript by means of two alternative strategies: the EMCV-IRES[33] or P2A-skipping ribosome sequence[34]. By interchanging the positions of E2A(DBP) and E4(Orf6) in the bicistronic transcriptional units, we generated four different Adenovirus Helper plasmids (named μHelper, pAμH1-4) about half the size of the original plasmid, as reported in Supplementary Fig. 8a. We compared these constructs by quantifying AAV production yield through quantitative PCR (qPCR). All μHelper plasmids led to AAV production, albeit to a lesser extent than the full-length Helper plasmid. Among these, the pAμH-3 plasmid (pCMV-E2A[DBP]-IRES-E4[Orf6]) exhibited the highest yields, as shown in Supplementary Fig. 8b. We attributed the lower production yield to the absence of the VaRNA-I ncRNA. Indeed, co-transfection of VaRNA-I along with puH-3 restored production efficacy (Supplementary Fig. 9).

To achieve inducible expression of the Helper genes using the CASwitch v.2 system, we introduced the direct repeat (DR) element into the 3′ untranslated region (UTR) of the E2A(DBP)-IRES-E4(Orf6) cassette and placed it downstream of pTRE3G (p3G-AμH3-DR), as depicted in Fig. 4d. We then qualitatively assessed the capability of the CASwitch v.2 system in controlling expression of helper genes for inducible AAV vector production in the context of transient triple transfection manufacturing and compared it to that of the state-of-the-art Tet-On3G system. Specifically, we employed EGFP as the transgene for generating AAV vectors, with fluorescence quantification in transduced cells serving as a qualitative indirect measure of production yields (Fig. 4e). We assessed production yields both in the presence and absence of doxycycline, providing a qualitative evaluation of the

Tet-On3G and the CASwitch v.2 systems' performance in AAV vector production, as reported in Fig. 4e-g. Infection results confirmed that when controlling Helper genes' expression with the Tet-On3G system, viral production occurred even in the absence of doxycycline, because of leaky expression of the viral Helper genes. Conversely, when controlling Helper genes with the CASwitch v.2 system, there was a significative reduction in AAV production in the absence of doxycycline, as measured by the percentage of infected cell, while maintaining high production yields in its presence. Despite viral production not being completely shut off, this proof-of-principle experiment shows that with proper fine-tuning, the CASwitch v.2 system could represent an effective solution to prevent unintended toxic viral gene expression, thus paving the way for the development of inducible AAV producer cell lines.

## Discussion

In this study, we applied a quantitative synthetic biology approach to overcome the performance limitations of current inducible gene expression systems thereby enhancing their reliability and increasing their potential applications. Through mathematical modelling, we engineered synthetic gene circuits that can overcome current trade-offs without changing the individual biological parts. We combined two well-known network motifs, the Coherent Feed-Forward Loop and the Mutual Inhibition, into a new Coherent Inhibitory Loop (CIL) circuit, that can act as a switch for gene expression. We hypothesised that the CRISPR-Cas13d endoribonuclease CasRx could facilitate mutual inhibition through a "sponge effect" arising from the irreversible binding of the CasRx to its DR sequence. While further experiments are necessary to conclusively demonstrate this sponging effect, the CasRx endoribonuclease enabled us to develop the CASwitch gene expression system with significant performance enhancements over the widely used Tet-On3G system.

To demonstrate the performance and versatility of CASwitch, we applied it to three different scenarios: enhancing a copper whole-cell biosensor, controlling expression of a toxic gene, and enabling inducible production of Adeno-Associated Viruses (AAVs). A copper biosensor is a device that can detect and measure the concentration of copper in a sample. Copper is an essential trace element for many biological processes, but it can also be toxic at high levels. Therefore, monitoring copper levels is crucial for environmental and biomedical applications. Tight control of toxic gene expression is encountered in the context of industrial biotechnological applications due to accumulation of the recombinant gene of interest, or to provide a kill switch to synthetically modified organisms. AAV vectors are tools that can deliver genes into cells for gene therapy or gene editing purposes. AAV vectors are derived from harmless viruses that can infect many types of cells without causing disease. However, producing AAV vectors is challenging and costly, requiring specialized facilities and equipment[24,25]. Therefore, developing methods to improve AAV vector production is vital for advancing gene therapy and gene editing technologies[26–30].

In all the scenarios, the CASwitch outperformed the Tet-On3G system in terms of precision, efficiency, and robustness. In the copper biosensor, the CASwitch effectively increased both sensitivity and dynamic range over its state-of-the-art counterpart, leading to more sensitive and accurate detection of copper levels. By swapping the metal responsive promoter with any other promoter of interest, the CASwitch v.2 can be transformed into a high-quality biosensor for any analyte. In the toxic gene scenario, we demonstrated that the CASwitch v.2 was able to maintain cell viability in the absence of doxycycline when controlling HSV-TK toxic gene expression in contrast to the classic Tet-On3G system. In the AAV vector production scenario, the CASwitch enabled a significant reduction of AAV production in the absence of doxycycline as compared to the Tet-On3G system, while retaining high level of production in its presence. Moreover, these

results suggest that the CASwitch could be used as a platform for the realization of inducible AAV producer cell lines to obtain a more controlled and scalable production of AAV vectors.

Although we proposed the CASwitch platform as a general strategy to enhance the performance of inducible gene expression system, we mostly relied on the Tet-On3G, since it is widely used in research and biotechnology and thus our results may benefit a large number of applications.

While we have defined general rules for tuning the mutual inhibition between CasRx and DR, we anticipate that further optimization of CasRx inhibition in the presence of the inducer molecule could enhance the effectiveness of this strategy. For example, we expect that post-transcriptional induced degradation strategies, such as aptamer-based translational control or degradation motifs, could offer a straightforward approach to increase CasRx inhibition. These strategies are likely to accommodate higher levels of CasRx and further reduce leakiness while retaining the same maximum induced expression.

Few studies have demonstrated the integration of transcriptional and post-transcriptional regulators to improve inducible gene expression in mammalian cells[4,15]. In these studies, combinations of bacterial transcriptional factor cascades, inhibitory shRNAs, or post-translational degradation tags were used to minimize basal gene expression, i.e. leakiness. However, the authors did not perform a comprehensive quantitative assessment of the system's performance, particularly regarding changes in the maximum expression. Additionally, these prior implementations involve many different components making their practical applications more cumbersome. Finally, they require optimization of inducer molecules' concentrations to achieve optimal fold change. More recently, DiAndreth et al.[16] developed a remarkable RNA-regulation platform consisting of nine CRISPR-specific endoRNases acting as RNA-level activators and repressors to construct logic functions and other circuits. However, they did not aim to improve inducible gene expression systems. Moreover, in their strategy, the authors exploited a different mechanism for post-transcriptional inhibition that relied on cleaving the DR sequence in the 5' UTR of the transcript.

In conclusion, here we show that CASwitch is a powerful platform for inducible gene expression, offering significant improvements over the current state-of-the-art systems. By using a quantitative approach based on synthetic gene circuits, this study has overcome some of the major challenges of inducible gene regulation, such as leakiness, maximal expression, and fold induction. By applying the CASwitch to three different scenarios, we demonstrated its versatility in biological research and biotechnological applications.

## Methods
### DNA cloning
All plasmids used in this study were constructed by Golden Gate cloning using the EMMA cloning platform[35] or NEBridge Golden Gate Assembly BsaI-HFv2 and BsmBI-v2 kits (NEB) and are listed in Supplementary Table 1. We used DNA parts provided with the EMMA cloning kit (Addgene kit #1000000119) and generated DNA fragments by PCR using Platinum™ SuperFi II Green PCR Master Mix (ThermoFisher) with in-house plasmids as templates. We amplified CasRx and DR(Rfx) sequences from pXR001 (Addgene # 109049) and pXR002 (Addgene # 109053), respectively. pTRE3G and rtTA3G sequences were amplified from pTRE3G-BI-ZsGreen1 (Takara #631339) and pCMV-Tet3G Vector (Takara # 631335). Instead, Gaussia and RedFirefly luciferases coding sequences were amplified from pCMV-Gaussia Luc vector (Thermofisher #16147) and pCMV-Red Firefly Luc Vector (Thermofisher # 16156). Finally, E2A(DBP), E4(Orf6), and VaRNAI sequences were took from the pHelper plasmid[36]. Repetition of four DRs was generated following a custom Gold Gate cloning protocol previously described[16]. Briefly, the repeated element (DR) carries

matching fusion sites at both ends. This leads to its concatenation in a Golden Gate cloning reaction. Introducing a second part with one fusion site matching the repeat element and the other matching the backbone seals the cloning construct. Stellar™ Competent Cells (Takara) were used for transformation, with 100 µg/mL kanamycin or ampicillin for selection. Bacterial colonies were screen by PCR-colony using MyTaq™ HS Red Mix (Meridian Bioscience). We verified positive individual clones' sequence by Sanger sequencing (Azenta Genewiz).

### Cell culture
We cultured HEK293T cells (ATCC, #CRL-3216) in DMEM high glucose GlutaMAX medium (Thermofisher) supplemented with 10% v/v Tet-Free fetal bovine serum (Euroclone) for all presented experiments, except for AAV production experiments where DMEM high glucose GlutaMAX medium was supplemented with 10% v/v fetal bovine serum (Euroclone). We cultured HeLa cells (ATCC, #CCL-2) in DMEM (Thermofisher) supplemented with 10% v/v Tet-Free fetal bovine serum (Euroclone) for all presented experiments.

### Luciferase assay
For luciferase assays, we used a reverse PEI transfection protocol. We seeded $5 \times 10^4$ HEK293T or $3 \times 10^4$ HeLa cells in each well of µCLEAR® Black 96 well plate (Greiner) and immediately transfected them with a mixture of DNA and PEI (MW 25000, Polysciences) (200 ng DNA/1 µl PEI in HEK293T or 200 ng DNA/0.8 µl PEI in HeLa; stock concentration 0.324 mg/ml, pH 7.5). We added doxycycline (0, 1, 2.5, 10, 100, 1000 ng/mL) or copper ($CuCl_2$; 0, 6.25, 12.5, 25, 37.5, 50, 75, 100 µM) to each well of the 96 well plate just before cell seeding. In all experiment comprising the CASwitch system, we transfected cells with rtTA3G: pTRE3G: CasRx encoding plasmids at a relative molar ratio of 1:5:1, except when we tested for higher relative amount of CasRx in the CASwitch v.1 system, where we used a 1:5:5 plasmid molar ratio. We measured Red Firefly luciferase and Gaussia luciferase expression using the Pierce™ Gaussia-Firefly Luciferase Dual Assay Kit (Thermofisher) on a Glomax Discovery plate reader (Promega). 48 hr after transfection, we lysed cells with 100 µl of 1 x lysis buffer. For each well, 20 µl of luciferase working solution was added using an automatic injector protocol. We calculated Luciferase Arbitrary Unit (Luciferase A.U.) by dividing each sample Gaussia luciferase expression by the constitutive Red Firefly luciferase expression determined from the same sample. We calculated Relative Luciferase A.U. by normalizing the set of Luciferase A.U. values by the mean of highest values assumed by Tet-On3G in experiments comparing CASwitch and Tet-On3G systems. We measured Firefly and Renilla luciferase activity using the dual luciferase assay (Promega) on a Glomax Discovery plate reader (Promega). 48 hr after transfection, we lysed cells with 100 µl of 1 x Passive lysis buffer and added reconstituted dual luciferase assay buffers following manufacturer automatic injector protocol. We calculated Luciferase Arbitrary Unit (Luciferase A.U.) by dividing each sample Firefly luciferase expression by the constitutive Renilla luciferase expression determined from the same sample.

### Cell viability assays
For luciferase-based cell viability assays, we used a reverse PEI transfection protocol. We seeded $5 \times 10^4$ HEK293T cells in each well of µCLEAR® Black 96 well plate (Greiner) and immediately transfected them with a mixture of DNA and PEI (MW 25000, Polysciences) (200 ng DNA/1 µl PEI; stock concentration 0.324 mg/ml, pH 7.5). We added doxycycline (0, 1000 ng/mL) and Ganciclovir sodium (GCV; 100 µg/mL) (MedChem Express; #HY-13637A) to each well of µCLEAR® Black 96 well plate (Greiner) just before cell seeding and transfection with the DNA/PEI mixture of plasmids of interest. 72 hr after transfection, we assessed cell viability via CellTiter Glo kit (Promega). We added 100 µL of Cell Titer-Glo Reagent, reconstituted following

manufacturer's instructions, in each well containing already 100 µL of growth medium. We mixed the entire 96-well plate for 2 min on luminometer built-in orbital shaker (200 rpm) and incubated the samples for 10 min at room temperature; then, we measured luminescence on a Glomax Discovery plate reader (Promega). For crystal violet (CV) staining, we coated six-well plates with poly-D-lysine (Gibco) following manufacturer's recommendations before seeding $4 \times 10^5$ HEK293T cells each well. 24 hr after cell seeding, we added doxycycline (0, 1000 ng/mL) and Ganciclovir sodium (GCV; 100 µg/mL) (MedChem Express; #HY-13637A) and immediately transfected them with a mixture of DNA and PEI (1500 ng DNA/8 µl PEI; stock concentration 0.324 mg/ml, pH 7.5). 72 hr after transfection, we washed cells three times with PBS and incubated with 1 ml of pure EtOH each well at -80C°for 24 hr to fix them. After fixation, we added 1 ml of CV solution (500 mg Cristal Violet, Sigma; 25% v/v Et-OH / H₂O) to each well and incubated for 15 min; then we washed cells three times with PBS.

## AAV vector production, extraction, and RT-qPCR titration in crude lysate

For AAV production, we seeded $4 \times 10^5$ HEK293T cells in each well of a 6-well plate and, after 24 hr, transfected them with a mixture of DNA and PEI (1500 ng DNA/8 µl PEI; stock concentration 0.324 mg/ml, pH 7.5). HEK293T cells were transfected with an equimolar amount of pAAV2.1-CMV-EGFP plasmid (pTransgene)[37], the pHAdV-5 helper plasmid containing the E2A, E4, and VA RNA helper genes (pHelper)[36] or the synthetic Helper plasmids (AµH Helper plasmids), and the pAAV2/2 packaging plasmid encoding the AAV serotype 2 Rep and Cap genes (pPackaging)[38]. To standardize the mass of transfected plasmids among transfection experimental conditions we added a filler pcDNA3.1 "empty" plasmid (Thermofisher). For inducible AAV vector production experiments, we added doxycycline (0, 1000 ng/mL) to each well of a 6 well plate just before transfection.

72 hr after transfection, we pelleted HEK293T cells at 1000 x g for 5' and resuspended them into 500 µl of AAV-Resuspension Buffer (50 mM Tris-HCL, pH 8; 200 mM NaCl; 2 mM MgCl₂). Then, we lysed resuspended cells by 4 cycles of freeze and thaw in a dry ice/ethanol bath and we clarified lysates by centrifugation at 18000xg for 10'; then, we treated lysates for RT-qPCR titration. We incubated 5 µl of clarified crude lysates containing rAAV particles with 50 units (U) of DNAse I (Roche) for 15 hr at 37 °C, then inactivated at 75 °C for 30 min. We diluted DNAse-treated samples with PCR-grade water to a volume of 40 µl and added 10 µl of proteinase K (> 600 mAU/mL) at 56 °C for 2 hr, then inactivated at 95 °C for 30 min. To generate standard curves for absolute qPCR we linearized pTransgene (p8-pAAV2.1 CMV eGFP3)[37] with XhoI and NheI enzymes (NEB) and diluted in PCR-grade water from $2.5 \times 10^{\wedge}8 - 25$ copies/µL in 10-fold serial dilutions. For each qPCR reaction we added 5 µl of sample (Standard or processed crude lysate) to reaction mix composed of 0.5 µM forward primer, 0.5 µM reverse primer, 12,5 ul of Light-Cycler® 480 SYBR Green I Master Mix (Roche), and PCR-grade water for a total reaction volume of 25 µL. We used primers (Eurofins) targeting the BGHpa in the pAAV2.1-CMV-EGFP linearized plasmid fragment; Forward Primer: 5′-GCCAGCCATCTGTTGT-3′; Reverse Primer: 5′-GGAGTGGCACCTTCCA-3′. We measured RT-qPCR reaction on a LightCycler® 96 Instrument (Roche) using the SYBR-green channel and with cycling conditions as follows: 95 °C for 10 min, followed by 40 cycles of 95 °C for 15 s and 60 °C for 1 min. We analysed RT-qPCR reaction using the LightCycler® 96 Software (Roche). We calculated viral genome concentration (Vg/mL) using the formula: Vg/mL = [(A/B) × 10 × 1000], where A is the interpolated total viral DNA amount in each well, B is the starting volume of the processed crude lysate sample (5uL), and 10 and 1000 are the dilution factors.

## Crude lysate transduction assays

We used 20 µL of crude lysate, obtained from HEK293T cells transfected to produced AAV vectors, to transduce $2.5 \times 10^4$ HEK293T cells in each well of a 96 well plate (Corning). We analysed percentage of GFP positive cells by flow cytometry. 48 hr after transduction, we harvested cells using trypsin-EDTA and washed three times with phosphate saline buffer (PBS). We kept cells in ice until analysis with Accuri C6 flow cytometer (BD). We determined cell autofluorescence threshold using samples transduced with crude lysate from cells transfected solely with pTransgene. We acquired at least $1 \times 10^4$ cells for each sample. We analysed acquired data using the Accuri C6 software (BD).

## Software and statistics

For many of the experiments, we chose a sample size of $n = 4$ (four experimental repeats) because it is standard practice in the field. We performed all key experiments at least twice; results were consistent across these replicates and the data presented in the article a representative of the trends we observed. We prepared graphs with GrapPad Prism 9.2.0 (http://www.graphpad.com). We used GraphPad Prism 9.2.0 also for statical purposes. Values are the means of at least four biological replicates ± standard deviation (s.d.). We assessed normal data distribution using D'Agostino & Pearson test. Two-way analysis of variance (ANOVA – one tailed) was used for statistical comparison of the data in the HSV-TK cell viability experiment and inducible AAV production.

## Mathematical Modelling and In Silico analyses

We developed deterministic Ordinary Differential Equation (ODE) models to qualitatively portray changes in the concentrations of the main biochemical species within the systems. The derivation of these equations was performed in accordance with the principles of the laws of mass-action and Michaelis-Menten kinetics. Additional details regarding the construction and examination of these models can be found in the Supplementary Note 1. (File: Supplementary Note 1)

## Deterministic simulations of the ODE model

All the simulations of the ODE model were run using MATLAB (v. 2022b). The ode15s function was used to numerically integrate the ODE model with zero initial conditions over a time span set to [0, 100] h. Hence, the steady-state value for each molecular species was taken at the end of a numerical simulation of 100 h as doxycycline concentration was changed. The parameter values used to simulate the ODE model were partially taken from literature and partially chosen by fitting the model with experimental data, as reported in Table 1 in Supplementary Note 1.

## Model fitting

The model fitting was performed using MATLAB (v. 2022b). In detail, fmincon function in combination with a multistart algorithm were used to find the model parameters that best fit the steady-state expression levels reported in Supplementary Note 1. The algorithm aimed ai minimizing the following loss function:

$$L(\theta) = \sum_{i=1}^{N} \left(Y_i - \hat{Y}_i(\theta)\right)^2$$

where $\theta$ is the vector of parameters to be estimated, $Y_i$ is the value experimentally collected, and $\hat{Y}_i(\theta)$ its prediction. First the TF parameters were fitted. Subsequently, we constrained the parameters that were common to both TF and MI before tuning the remaining parameters of the MI. At the same way, we fitted the last parameters of CIL. A non-linear restriction was incorporated into the algorithm to replicate the observed leakiness values of the systems during simulations.

## Analytical characterization of the fold change activation of the systems

To characterize the fold change activation ability of the systems, we exploited the analytical solutions of the ODE models reported in Supplementary Note 1. The expressions were derived by mean of Wolfram Mathematica using the command Solve and manually reshaped. We compared each other by means of the command Reduce and some constrains added to match biological limitations.

## Reporting summary

Further information on research design is available in the Nature Portfolio Reporting Summary linked to this article.

## Data availability

Sequences for all plasmids used in this study are provided as GenBank files in Source Data. All data required to support the findings of this study in the study can be found in the paper and/or the supplementary information files. Source data accompanying this manuscript include measured Luciferase activities, fold-changes, cell vitality measurements, and AAV production yield via RT-qPCR and/or transduction assay. The raw data are available from the corresponding authors upon request. Source data are provided with this paper.

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

## Acknowledgements

We are grateful to Alberto Auricchio for sharing his expertise on AAV, and to Monica Doria and Barbara Tumaini for technical help. This work was supported by University of Naples Federico II and Compagnia di San Paolo - Programme STAR Plus, by Fondazione Telethon, and in part by the EU H2020 project iPC 826121.

## Author contributions

G.D.C. and D.d.B. designed the research; G.D.C. designed, built, and experimentally validated circuits and carried out their application experiments; V.F. helped to perform experiments and carried out the mathematical modelling and simulations. G.D.C and D.d.B. performed data analysis; G.D.C and D.d.B wrote the paper. V.F. wrote supplementary notes. All authors contributed to review and editing; D.d.B. supervised the project and secured funding.

## Competing interests

The authors declare no competing interests.
