## [Peer Review File · Nature Communications]

Reviewers' Comments:

Reviewer #1:

Remarks to the Author:

The manuscript by De Carluccio et al., details the construction of a genetic circuit for target gene induction in HEK cells with improved dynamic features.

In particular, noting that available state of the art system Tet3G has a relatively high leakiness, the authors aimed at reducing the basal leakiness without the inducer, without compromising the activated levels with maximal inducer.

To do so, the authors first performed computational simulations of different architectures; then implement the different variants with a CasRx endoribonuclease-based system and compare them to the tet3G system; finally deploy the best performers in 3 selected application of biotechnology interest.

Overall the paper is targeting an area of high interest (induction of target genes), where innovation has kind of stagnated in the last few years. It is hard to overstate how important would be for biotechnology as well as research to have an inducible gene expression system with improved performance. The results are strongly supporting the claim that the CASwitch has reduced leakiness, and in the CIL, the max-induced values are high. The results would be of interest of a broad audience, and I expect it could be implemented rapidly in the community.

Other than some specific comments (see below) I have 4 main line of critiques in order of relevance:

1. is the lack of exploration of what different doses of plasmids in the transient transfection of the circuit components effect would have on the induction of the target gene. Different amount of plasmids can affect basal and induced levels of gene expression; it would be important to make sure that these changes that one can achieve just by manipulating amount of plasmids in their transfection reactions, are not producing changes that are helpful, hence supporting the need for an integrated circuit.

2. Starting from Fig.2 , fold gain looks impressive and is backed up by extensive data. In the text you are claiming that this is due to the implementation of circuit architecture based on mutual inhibition of Z and Y. I am not convinced that this is proven with the current dataset. I want to note here that this may not be too too important important for the applications. But if it is a main claim for the paper it would need to be better substantiated (or viceversa, rewriting the text with less emphasis on your circuit implementing exactly that architecture).
e.g. if there is mutual inhibition, then having the gLuc with 4xDR should have more inhibition of Z to Y; what would be your prediction of the circuit behavior? Can you test it experimentally?

3. The claims are often general, whereas I see only data presented for HEK cells; if that is the case, it should be stated clearly that these results apply to HEK cells only (so far). Different cell lines have different signal-to-noise ratio of different promoters, and these results could be very different in other cells lines (or not!, but at this point we don't know).

4. I think the manuscript readability and impact would improve if there was a quantitative definition of most of the terms used. There is abundant use of terms like "leakiness" or "high maximal expression", but I could not find formal definitions. As examples: lines 54-55, 162, ... and see also other examples in the specific comments below.

Specific comments:

Fig. 2 and 3 of the Note 1 seem very relevant; I would encourage the authors to consider ways to incorporate a distillate of them in the main text/figures. The main message being the behavior of the system when some key parameters change, which is very interesting to know, and would go

hand-in-hand with the analysis of what different doses of plasmids would do to the in vitro system.

Line 73, what performance improvements do you recognize? May be good to spell them out first. How do you define performance?

Line 77-81: the description of the results shown in figures is confusing; cannot readily recognize what you are claiming here. Where are you looking at when you say things like "thus enabling a further increase in the maximal expression of Z"? When you compare MI and CIL networks, how are you comparing them? Just by looking at them? Or are you using some kinds of metrics, either qualitative or quantitative?

Conclusion 81-83 is not fully justified by data, main challenge is that "reduction in leakiness without compromising the maximal achievable transcription rate" is not well defined. Do you have certain thresholds in mind to say that leakiness is "reduced"?

Starting from Fig. 2c becomes relevant to know exactly how you did the genetic engineering; consider moving that high level description in the results from the methods. It seems to me that it is still transient transfection, is that the case? If that is the case, does the amount of plasmids that you are using make a difference?

Fig. 2c-l: not clear why you chose here a gLuc with only one CasRX Direct Repeat loop? Do you still think it implements mutual inhibition? Where is the sponging coming from?

Fig. 2d and e; took me a while to understand they are the same thing, only one is log one linear; may consider keeping only one (log?). How long of an induction are we talking about here? What about in the computational system?

Supplem Fig. 2-4: great controls!!

l.162 Same suppression of leakiness: how do we know that? What do you define as "suppression of leakiness"? Is it a particular arbitrary unit of luciferase? Or a fraction of the 0 Dox of the TetOn3G?

Supplem Fig. 5, fold induction is in log here, whereas in main text is in linear, makes it confusing and hard to compare.

After Fig.2 , it looks impressive the fold gain; I am not sure it is implementing the circuits that you think it is, which may not be too important for the applications. But if it is a main claim would need to be better substantiated.

e.g. if there is mutual inhibition, then having the gLuc with 4xDR should have more inhibition of Z to Y; what would be your prediction of the circuit behavior? Can you test it experimentally?

Line 213. Made me think: is this important? Would this improvement make a difference in terms of biosensors? How? We are able to pick up certain dosage differences that are known to be difficult to recognize?

Fig. 4a-c, cool application, easy to see how it could be helpful!

Reviewer #2:

Remarks to the Author:

This manuscript presents an innovative approach to improving the dynamic range of inducible gene expression systems. The approach developed by the authors addresses a common challenge in the design of genetic circuits – the leakiness of gene expression – while maintaining high levels of maximum expression. Specifically, the authors achieve this goal using post-transcriptional control mediated by CRISPR-Cas endoribonuclease (CasRx). Computational modeling is used to evaluate three candidate genetic circuits with the goal of minimizing leakiness and maximizing gene expression upon induction. Ultimately, a gene circuit that combines a feed-forward loop and

mutual inhibition is selected as the best candidate (CASwitch V.2). In this circuit, CRISPR-Cas endonuclease (CasRx) is used to implement the mutual inhibition between output signal protein and CasRX through posttranscriptional degradation of mRNA, which leads to a significantly reduced leakiness without affecting the maximum output expression levels. The authors demonstrate experimentally the behavior of CASwitch V.2 in HEK293T cells. Their results support the use of CASwitch V.2 in applications such as whole-cell sensors, toxic gene expression, and viral vector production. The following issues were identified:

- In line 23 - (TF) is introduced as a transcription factor. Further in the manuscript, lines 66 and 592, TF is used also to indicate the naïve configuration of an inducible gene expression system. The authors should be more precise to avoid confusion or misinterpretations.
- lines 167 to 173 – The authors describe an experiment to test the robustness of the CASwitch V.2 circuit by using alternative promoters for the expression of the main transcription regulator. Despite this component (promoter) being key for the behavior of CASwitch V.2, more convincing evidence is needed to demonstrate the robustness of the circuit. A better experiment to test the robustness of the genetic circuit would be to evaluate the circuit's behavior in other cell lines where circuit components behave differently in terms of kinetic parameters.
- The authors describe viral production using this CASwitch V.2. one of the main issues associated with viral production is cell viability/survival. Such metrics (i.e., cell survival/viability) after viral production should be reported.
- The experiments based on the use of transduced cells to measure the virus production are do not include assessment of variability and statistical significance. makes the result hard to understand since it does not indicate statistical significance. The authors should include an alternative (additional) readout such as qPCR measurements.
- A discussion of the CASwitch system's behavior in comparison to their circuits that share a similar topology for mutual inhibition but employ different types of controls, such as transcriptional-based or posttranslational-based control (with citations) should be included to support the authors claims of superior and innovative results obtained with the CASwitch system.

Response to Reviewers

In what follows we replied to reviewers' comments and suggestions. We renumbered the comments to make cross-reference easier and we formatted in *italic* the reviewers' comments, whereas our replies are in normal text. We also highlighted changes made in the revised manuscript to make it easier for the reviewers to detect them. All line and page numbers refer to the revised manuscript file with tracked changes.

Reply to Reviewer #1:

Overall, the paper is targeting an area of high interest (induction of target genes), where innovation has kind of stagnated in the last few years. It is hard to overstate how important would be for biotechnology as well as research to have an inducible gene expression system with improved performance. The results are strongly supporting the claim that the CASwitch has reduced leakiness, and in the CIL, the max-induced values are high. The results would be of interest of a broad audience, and I expect it could be implemented rapidly in the community.

We really thank the reviewer for his/her comments and for appreciating our work.

1.1. *[one critique] Is the lack of exploration of what different doses of plasmids in the transient transfection of the circuit components effect would have on the induction of the target gene. Different amounts of plasmids can affect basal and induced levels of gene expression; it would be important to make sure that these changes that one can achieve just by manipulating amounts of plasmids in their transfection reactions, are not producing changes that are helpful, hence supporting the need for an integrated circuit.*

We agree with the reviewer's observations, indeed in our original manuscript, we used a plasmid molar ratio of 1:5 between rtTA3G and pTRE3G encoding plasmids, as this is what was recommended in the Tet-ON3G system's protocol provided by Takara Inc. However, to comply with reviewer's comment, in the revised manuscript, we have now explored also a 1:1 molar plasmid ratio between the rtTA3G and pTRE3G plasmids and compared it to the ratio of 1:5 that we used in our original manuscript. Specifically, we have now performed an additional experiment, whose results are reported in the **new Supplementary Figure 5**, in which we compared: (i) the state of the art Tet-ON3G system when using a plasmid molar ratio between the two plasmids (pCMV-rtTA3G : pTRE3G-fLuc-DR) of 1:5 and of 1:1; and (ii) the CASwitch v.2 with molar ratio among the three plasmids (pCMV-rtTA3G : pTRE3G-fLuc-DR : pCMV/TO-CasRx) of 1:5:1 and of 1:1:0.2 (observe that we kept the ratio between fLuc and CasRx fixed to 5 to 1, this is why when the ratio of CasRx is indicated as 0.2, i.e. CasRx is 5 times less than rtTA3G) (**Supplementary Figure 5a**). As shown in the **revised Supplementary Figure 5b**, for the state-of-the-art Tet-ON3G system, when transfecting HEK293T cells with the rtT3G and pTRE3G plasmids at 1:1 molar ratio (green line), the result is a lower leakiness as compared to the 1:5 ratio (yellow line), but also lower maximum response. In turn, this yields only a moderate increase in the fold induction levels corresponding to 4.7 times the one obtained with the Tet-On3G 1:5, as reported in the revised **Supplementary Figure 5c**. On the contrary, as reported in **revised Supplementary Figure 5b-c**, it can be appreciated that the CASwitch v.2 (red and blue lines) is robust to changes in the plasmid molar ratio, in terms of low leakiness, high maximum expression, and high fold induction levels. These results demonstrate that the performance of the state-of-the-art Tet-ON3G system cannot be improved just by manipulating the relative molar amounts of plasmids in the

transfection reactions. We have now modified the main text to incorporate these new results (page 4 - lines 176-181).

1.2. Starting from Fig.2, fold gain looks impressive and is backed up by extensive data. In the text you are claiming that this is due to the implementation of circuit architecture based on mutual inhibition of Z and Y. I am not convinced that this is proven with the current dataset. I want to note here that this may not be too important for the applications. But if it is a main claim for the paper it would need to be better substantiated (or viceversa, rewriting the text with less emphasis on your circuit implementing exactly that architecture) .e.g. if there is mutual inhibition, then having the gLuc with 4xDR should have more inhibition of Z to Y; what would be your prediction of the circuit behavior? Can you test it experimentally?

We value the reviewer's insights into our systems and its recommendations regarding the mutual inhibition architecture. We anticipate that by introducing the gLuc_4xDR in the CASwitch, this would lead to increased leakiness in the absence of doxycycline, because of a stronger sponging of CasRx by the gLuc_4xDR, while the maximum expression should be unaffected. Hence, the fold induction of the CASwitch with gLuc_4xDRs should be lower than the one achieved with gLuc_1xDR. However, we believe that this experiment would still not conclusively demonstrate that mutual inhibition results from a sponging effect of the DR on the CasRx, as this would require direct measurements of the binding between the CasRx protein and the gLuc mRNA, which would require extensive efforts that are out of the scope of this manuscript. For this reason, to comply with the reviewer's suggestion, we decided to revise the manuscript to emphasize that the CASwitch behavior is consistent with a mutual inhibition architecture albeit we cannot prove it definitively (page 3 - lines 106-108, 110-114, 120-122; page 8 – lines 312-316). We also modified Figure 2,3,4 by replacing the blunted arrow from the DR to the CasRx with a dashed line to make it clear to the reader that this is only a hypothetical regulation.

1.3. The claims are often general, whereas I see only data presented for HEK cells; if that is the case, it should be stated clearly that these results apply to HEK cells only (so far). Different cell lines have different signal-to-noise ratio of different promoters, and these results could be very different in other cells lines (or not!, but at this point we don't know).

We agree that synthetic circuits may behave differently in other cell lines. To address this point, we have now performed additional experiments to test CASwitch v.2 circuit and the state of the Tet-On3G system also in HeLa cells. The results are now reported in a new Supplementary Figure 6 and described in page 4 - line 182-184 and show that the CASwitch v.2 works as expected also in this cell line.

1.4. I think the manuscript readability and impact would improve if there was a quantitative definition of most of the terms used. There is abundant use of terms like "leakiness" or "high maximal expression", but I could not find formal definitions. As examples: lines 54-55, 162, ... and see also other examples in the specific comments below.

To comply with reviewer's suggestions, we have now revised Figure 1 and its caption to graphically illustrate the definitions of the terms in a new panel (a). We also revised the text in the main manuscript (page 2 - lines 39-44, 76-77)

1.5. Fig. 2 and 3 of Note 1 seem very relevant; I would encourage the authors to consider ways to incorporate a distillate of them in the main text/figures. The main message being the behavior of the

system when some key parameters change, which is very interesting to know, and would go hand-in-hand with the analysis of what different doses of plasmids would do to the in vitro system.

We thank the reviewer for this suggestion. We have now added **an extra panel (f) to the revised Figure 1**, to summarize the robustness results reported in Note 1 and we have modified the main text to describe this new panel **(page 3 - line 85-89)**

1.6. *Line 73, what performance improvements do you recognize? May be good to spell them out first. How do you define performance?*

We have now revised the manuscript to emphasize that the three features that define the performance are “Leakiness”, “Maximum expression,” and “Fold induction”, and hence an improvement in the performance relative to the naïve configuration means a reduced leakiness, an unchanged or increased maximum expression, and increased fold-induction. We have now modified the main text to make this point clearer **(page 2 - lines 39-44, 76-77)**. Please also refer to point 1.4.

1.7. *Line 77-81: the description of the results shown in figures is confusing; cannot readily recognize what you are claiming here. Where are you looking at when you say things like “thus enabling a further increase in the maximal expression of Z”? When you compare MI and CIL networks, how are you comparing them? Just by looking at them? Or are you using some kinds of metrics, either qualitative or quantitative?*

To make the results clearer, we have now revised both the text **(page 2 - lines 39-44, 76-77, 83-85)** and modified **Figure 1** and its caption. In the revised manuscript, we have now made it clear that the comparison is always against the naïve configuration consisting of the transcription factor X driving expression of the target Z (please observe that in the previous version of the manuscript, this configuration was labelled “TF” but to avoid any confusion, as raised by Reviewer 2 in point 2.1, we have now renamed it “NC” for Naïve Configuration). In the **revised Figure 1**, we have now: (i) added a new panel (a) with the definition of the three features; (ii) added labels to the panels (c), (d), and (e) to highlight the fact that these plots represents the three features for each circuit configuration; (iii) added colored open circles to the plots in panels (c), (d), and (e) to guide the reader and to highlight the difference between the NC and the three genetic circuits (CFFL4, MI and CIL); and (iv) we modified the caption of Figure 1 to formally define the three features referring to the new panel (a).

1.8. *Conclusion 81-83 is not fully justified by data, main challenge is that “reduction in leakiness without compromising the maximal achievable transcription rate” is not well defined. Do you have certain thresholds in mind to say that leakiness is “reduced”?*

We apologize for having caused confusion; we wrongly wrote “maximal achievable transcription rate” rather than “maximum expression”. As reported in our reply to points 1.6 and 1.7, we have now clearly defined the features we are referring to in the **revised Figure 1a** (i.e. leakiness, maximum expression and Fold induction) and also better explained that we are comparing the circuits against the Naïve Configuration (NC), therefore what we meant by “reduction in leakiness without compromising the [maximum expression]” was that the leakiness of the genetic circuits is smaller than the one of the Naïve Configuration, whereas the maximum expression is close to the one of the Naïve Configuration. We have updated the main text on **page 2 - lines 39-44, 78-82** to explain these points.

1.9. Starting from Fig. 2c becomes relevant to know exactly how you did the genetic engineering; consider moving that high level description in the results from the methods. It seems to me that it is still transient transfection, is that the case? If that is the case, does the amount of plasmids that you are using make a difference?

The reviewer is correct, all the experiments were done by transient transfection of HEK293T cells. We have now modified the main text (page 3, lines 130-133) to clearly state this fact and moved part of the description from the Methods to the main text as suggested. Regarding the amount of plasmids, since each circuit is made up of three plasmids (except for the Naïve Configuration that has only two plasmids) what affects the performance of the circuits is the relative concentration of the plasmids, which we have now better explored to comply with the reviewer comment in point 1.1, and described in our reply to that point.

1.10. Fig. 2c-l: not clear why you chose here a gLuc with only one CasRX Direct Repeat loop? Do you still think it implements mutual inhibition? Where is the sponging coming from?

We opted for a single direct repeat (DR) because of its greater effectiveness in leading to the CasRx-mediated repression of the target gene harboring the DR (gLuc_1xDR), as evidenced by our results reported in Figure 2b. This effectiveness is important to maximally reduce leakiness. We do believe that even one direct repeat still implements a mutual inhibition as ideally each CasRx molecule will cleave and then irreversibly bind the DR. This irreversible binding is what should create the sponging effect even with one DR per gLuc; imagine you have an excess of gLuc_1xDR mRNAs over CasRx proteins, then all the CasRx proteins will be inhibited as each CasRx protein will cleave the single DR present in one gLuc mRNA, but then it will become inactivate because of the irreversible binding to the DR, thus preventing it to bind additional DRs present in the other gLuc mRNAs. Therefore, the gLuc will effectively sponge out the CasRx proteins with a 1:1 stoichiometry.

1.11. Fig. 2d and e; took me a while to understand they are the same thing, only one is log one linear; may consider keeping only one (log?).

We do appreciate reviewer feedback to improve the clarity of our manuscript. We'd like to retain both log and linear scales to better illustrate the difference in terms of leakiness, which can be appreciated only using the log-scale, and maximum response that can be appreciated only using the linear scale. Thus, to address this point, we have now revised Figure 2 and its caption to make this point clearer by reorganizing the panels and by clearly indicating log and linear scale on the Y axis of each plot.

1.12 How long of an induction are we talking about here? What about in the computational system?

Each experiment has been performed in transient transfection providing doxycycline at the time of transfection and acquiring results 48hr after transfection, except for HSV-TK viability and AAV experiments in Figure 4 that required 72hr of incubation. This timing is consistent with the mathematical model where the dominant rate parameter (i.e. the smallest) reported in Table 1 of the Supplementary Note has an order of magnitude of about 0.1 h^{-1} corresponding to a time constant of about 10 h (i.e. $1/0.1$). We modified the Methods section (page 10 – lines 409-411, 416, 432-434; page 11 - 456-458) to make this point clearer.

1.13. *Line 162 - Same suppression of leakiness: how do we know that? What do you define as “suppression of leakiness”? Is it a particular arbitrary unit of luciferase? Or a fraction of the 0 Dox of the TetOn3G?*

We thank the reviewer for pointing this out. The answer is “a fraction of the 0 Dox of the TetOn3G”. Indeed, in the text in the original manuscript, we referred to the original Figure 2h to illustrate the comparative performance between the standard Tet-On3G system and our novel CASwitch. In response to the reviewer’s comment, we have now **revised Figure 2, specifically in panels d and g**, to explicitly illustrate the extent of leakiness, which we have now defined in the revised Figure 1a as the luciferase observed at 0 ng/mL of doxycycline – a condition where no luciferase expression is anticipated. The term ‘suppression of leakiness’ refers to the reduced luciferase activity in the CASwitch system in the absence of doxycycline, relative to that of the Tet-On3G system. Moreover, for clearer interpretation, in **revised Figure 2**, we also changed the *y-axis* label in panels d and g from ‘Relative luciferase A.U.’ to ‘Relative luciferase (%)’. This label better reflects the fact that the values represent the luciferase level as a percentage of the maximum expression obtained by the Tet-On3G system at 1000 ng/mL doxycycline. Hence, a value of 100 implies equivalent luciferase level to that of the maximum expression of the Tet-On3G system, whereas a value of 1 represents a luciferase level equal to 1% of the maximum expression of the Tet-On3G system. We have now amended the manuscript to incorporate these changes **(page 4, lines 168-170)**.

1.14. *Supplem Fig. 5, fold induction is in log here, whereas in main text is in linear, makes it confusing and hard to compare.*

We have now reverted to a linear scale to make comparison easier. Please note that the original Suppl. Figure 5 has now become Suppl. Figure 7 in the revised manuscript.

1.15. *After Fig.2, it looks impressive the fold gain; I am not sure it is implementing the circuits that you think it is, which may not be too important for the applications. But if it is a main claim would need to be better substantiated. e.g. if there is mutual inhibition, then having the gLuc with 4xDR should have more inhibition of Z to Y; what would be your prediction of the circuit behavior? Can you test it experimentally?*

We agree with the reviewer’s comment. Indeed, as we also addressed in our reply to the reviewer’s points 1.2 and 1.10, we do expect that the introduction of CASwitch v.2 with a gLuc carrying 4xDRs would result in an inducible system with lower performance because of increased sponging thus reducing the function of the CasRx, as shown in Figure 2b. However, as this additional experiment would not definitively establish that mutual inhibition indeed results from a sponging effect of the DR on the CasRx and recognizing that an experimental exploration of this phenomenon falls beyond the scope of our current paper, we have opted to revise the manuscript and the Figures. Specifically, in the revised text of the Results and Discussion sections **(page 3 - lines 106-108, 110-114, 120-122; page 8 – lines 312-315)** we have now emphasized that mutual inhibition may be at play but that additional experiments should be carried out to prove this mechanistically. In addition, in Figures 1 to 4, we have now changed the flat arrow representing inhibition from Z to Y in the circuit diagrams with a dashed line, to highlight that this is hypothetical regulation and not definitely proven. We believe that these changes will make it clear to the reader that mutual inhibition is only a credible hypothesis but is not proven.

1.16 *Line 213. Made me think: is this important? Would this improvement make a difference in terms of biosensors? How? We are able to pick up certain dosage differences that are known to be difficult to recognize?*

We value the reviewer's input aimed at enhancing the readability of our manuscript. In the context of biosensors, a higher fold induction level is directly linked to the increased signal-to-noise ratio thus increasing the reliability of the biosensor. Additionally, a broader dynamic range, i.e. the fold change (defined as the ratio between the maximum response and leakiness), enables the discrimination of more analyte concentrations. In the original biosensor, lower concentrations of copper cannot be effectively distinguished, and the biosensor is activated only at very high copper concentrations, such as 100 μM . On the other hand, the implementation of our CASwitch v.2 allows for reliable activation at a lower copper concentration of 25 μM . Consequently, in response to the reviewer's suggestion, we have incorporated this comment in the revised manuscript **(page 5, line 233-235)**.

Reply to Reviewer #2:

This manuscript presents an innovative approach to improving the dynamic range of inducible gene expression systems [...] Their results support the use of CASwitch V.2 in applications such as whole-cell sensors, toxic gene expression, and viral vector production.

We thank the reviewer for the encouraging comments.

2.1. *In line 23 - (TF) is introduced as a transcription factor. Further in the manuscript, lines 66 and 592, TF is used also to indicate the naïve configuration of an inducible gene expression system. The authors should be more precise to avoid confusion or misinterpretations.*

We are grateful to the reviewer for pointing out this source of confusion in the manuscript. We have now modified the text and Figures throughout the revised manuscript by substituting the “TF” abbreviation for the naïve configuration, with the abbreviation “NC” (standing for Naïve Configuration)

2.2. *Lines 167 to 173 – The authors describe an experiment to test the robustness of the CASwitch V.2 circuit by using alternative promoters for the expression of the main transcription regulator. Despite this component (promoter) being key for the behavior of CASwitch V.2, more convincing evidence is needed to demonstrate the robustness of the circuit. A better experiment to test the robustness of the genetic circuit would be to evaluate the circuit’s behavior in other cell lines where circuit components behave differently in terms of kinetic parameters.*

To comply with the reviewer’s suggestion, we have now tested the CASwitch v.2 also in HeLa cells, whose results are now reported in the **new Supplementary Figure 6** of the revised manuscript. In HeLa cells, CASwitch v.2 reduces leaky gene expression by at least an order of magnitude compared to the Tet-On3G system, while maintaining high maximum expression levels, hence similar to its performance in HEK293T cells. As in the case of HEK293T cells, the decreased leakiness and the retention of high maximum expression led to a substantial gain in fold-induction of over 1 order of magnitude. These data demonstrate the robustness of CASwitch v.2 to enhance the performance of transcriptional inducible gene systems, also in HeLa, and support its broader applicability. We have now described this result in the main text (**page 4, lines 182-184**).

2.3. *The authors describe viral production using this CASwitch V.2. one of the main issues associated with viral production is cell viability/survival. Such metrics (i.e., cell survival/viability) after viral production should be reported.*

We agree with the reviewer that these metrics would be essential in the context of a stable inducible AAV producer cell line to show that in the absence of inducer (i.e. doxycycline) cells are indeed viable over the long term. However, in the manuscript we only presented a proof-of-principle application using transient transfection of HEK293T for AAV viral vector production, but we did not generate a stable cell line, as this would have required a lengthy and costly process that was out of the scope of the manuscript. In the context of transient transfection, cell viability/survival is not a concern because cells do not harbor any toxic viral genes before transfection, and they are lysed to extract AAV vectors after three days. To make this point clearer and avoid any confusion, we have now modified the text of this section in the revised

manuscript to clearly state that this is only a proof of principle, and more work is needed to prove its practical feasibility (page 6 - lines 263-264, 268-269, page 6-7 - 288-293). We hope that the reviewer will agree with us on this point.

2.4 The experiments based on the use of transduced cells to measure the virus production are do not include assessment of variability and statistical significance. makes the result hard to understand since it does not indicate statistical significance. The authors should include an alternative (additional) readout such as qPCR measurements.

We thank the review for pointing this out and we apologize if in our original Figure 4 did not show the assessment of variability and statistical significance for the AAV experiment. In response to the reviewer's valuable suggestion, we have now incorporated the evaluation of variability and statistical significance into **the revised Figure 4g** and modified its caption accordingly. We agree with the reviewer on the value of qPCR measurements for precise quantitative assessment of AAV production, however we would like to point out that these measurements would have required purification of AAVs from the crude lysates, which can be efficiently performed only at large-scale by transfecting a large number of cells (CELL STACK, Corning). However, in our manuscript we performed all the experiments of AAV production on a small-scale (6-well plate), as we needed to compare multiple conditions and constructs and large-scale production followed by purification would have been time and cost prohibitive. Therefore, considering that we aimed only to showcase AAV production with the CASwitch v.2 as a proof of principle application, we believe that the semi-quantitative approach based on crude lysates to transduce cells to compare AAV production between the Tet-On3G system and the CASwitch suffices. Nonetheless, we acknowledge the limitation of our qualitative assessment of AAV vector production, and we now amended the text in the revised manuscript to make the reader aware of this limitation as detailed also in our reply to point 2.3 (page 6 - lines 263-264, 268-269, page 6-7 - 288-293).

2.5 A discussion of the CASwitch system's behavior in comparison to the circuits that share a similar topology for mutual inhibition but employ different types of controls, such as transcriptional-based or posttranslational-based control (with citations) should be included to support the authors claims of superior and innovative results obtained with the CASwitch system.

We thank the reviewer for the feedback aimed to improve our manuscript. To comply with reviewer suggestion, we have now expanded the Discussion section in the revised manuscript to discuss the CASwitch in the context of the existing literature (page 9, lines 355-367).

Reviewers' Comments:

Reviewer #1:

Remarks to the Author:

My comments have been thoroughly addressed in the revised manuscript, I have no further question/comment.

Reviewer #2:

Remarks to the Author:

the authors addressed all the concerns raised by this reviewer.